# Vascularized Microfluidics and the Blood–Endothelium Interface

**DOI:** 10.3390/mi11010018

**Published:** 2019-12-23

**Authors:** Christopher A. Hesh, Yongzhi Qiu, Wilbur A. Lam

**Affiliations:** 1Department of Radiology & Imaging Sciences, Emory University School of Medicine, Atlanta, GA 30322, USA; chesh@emory.edu; 2Wallace H. Coulter Department of Biomedical Engineering, Georgia Institute of Technology and Emory University, Atlanta, GA 30322, USA; 3Department of Pediatrics, Division of Pediatric Hematology/Oncology, Aflac Cancer Center and Blood Disorders Service of Children’s Healthcare of Atlanta, Emory University School of Medicine, Atlanta, GA 30322, USA; 4Winship Cancer Institute of Emory University, Atlanta, GA 30322, USA; 5Parker H. Petit Institute of Bioengineering and Bioscience, Georgia Institute of Technology, Atlanta, GA 30322, USA

**Keywords:** blood vessel, microvasculature, endothelium, lab-on-chip, microfluidics

## Abstract

The microvasculature is the primary conduit through which the human body transmits oxygen, nutrients, and other biological information to its peripheral tissues. It does this through bidirectional communication between the blood, consisting of plasma and non-adherent cells, and the microvascular endothelium. Current understanding of this blood–endothelium interface has been predominantly derived from a combination of reductionist two-dimensional in vitro models and biologically complex in vivo animal models, both of which recapitulate the human microvasculature to varying but limited degrees. In an effort to address these limitations, vascularized microfluidics have become a platform of increasing importance as a consequence of their ability to isolate biologically complex phenomena while also recapitulating biochemical and biophysical behaviors known to be important to the function of the blood–endothelium interface. In this review, we discuss the basic principles of vascularized microfluidic fabrication, the contribution this platform has made to our understanding of the blood–endothelium interface in both homeostasis and disease, the limitations and challenges of these vascularized microfluidics for studying this interface, and how these inform future directions.

## 1. Introduction

Blood is a connective tissue composed of blood cells and plasma and serves a variety of vital homeostatic roles in the human body. These roles are mediated through biochemical and biophysical communication with the vascular endothelium in vessels that range widely in size and structure. Due to the limited distance of substance diffusion in tissues almost every cell in the human body lives within 100–150 μm of vascular endothelium [1]. In these far reaches, the vascular system serves to keep blood fluid [2], regulate the perfusion of end organs [3], prevent leukocyte (WBC) activation key in the immune [4] and inflammatory [5,6] responses, and regulate the delivery of macromolecules [7]. At specific organotypic microvascular networks, such as the brain [8], liver [9], and adipose tissue [10], additional niche homeostatic functions are assumed, including endothelium acting as a gatekeeper for organ development and tissue regeneration [11].

This is made possible through complex and highly regulated communication between the blood and endothelial cells (ECs) mediated by various adhesion mechanisms [12], cellular mechanical properties [13], and chemoattraction [14], as well as the fluid stresses of hemodynamics [15]. When this delicate balance is altered, disease results. Indeed, the pathology of the vascular system as a whole, as well as specific organotypic blood–endothelium interactions, have been implicated in ailments throughout the human body, including chronic liver disease [16], cystic fibrosis [17], pulmonary arterial hypertension [18], and sickle cell disease [19].

Our current understanding of the blood-vascular endothelium interface is predominantly based upon research using two model types. In vivo assays, the first model type, are the gold standard. In the study of vascular biology, these assays consist mostly of small animals. These models, which uniquely allow for the investigation of the blood–endothelium interface in a complete and functional mammalian physiology—and through this have provided much of our foundational understanding of human microvasculature—are nonetheless limited by their complexity. Specifically, quantification is made difficult by the inability to tightly control the biochemical and biophysical parameters of living systems. Furthermore, the imaging of cellular level events in these models can be technically challenging. Finally, while small animals recapitulate many important aspects of human vascular biology, its phenotype does not always reflect that of humans. This has led to conflicting results when comparing results from animal experimentation with what is seen in patients. For instance, mouse and humans exposed to various stimulants of the inflammatory cascade have gene expression profiles that show no correlation [20]. Attempts are made to address these limitations through the development of the second model type, reductionist in vitro assays, such as traditional cell cultures of primary human cells. These models, however, have been relatively constrained by their lack of complexity, such as a lack of physiologic fluid flow [21]. These models, and others such as parallel-plate chambers and cone-plate rotational viscometer systems which have better recapitulated in vivo features of fluid flow, nonetheless have failed to recapitulate many aspects of in vivo models.

In recent years, however, in vitro models have expanded in complexity, integrating important aspects of their in vivo counterparts. Among these are microfluidic models with endothelialized microchannels, which allow for the recapitulation of in vivo endothelial barrier function and scrutiny of blood–endothelium interactions in a more physiologic microenvironment than previously available. These platforms provide tight control of the geometry, as well as inputs such as media or whole blood, and the shear to which they expose the endothelium. Furthermore, with the use of transparent substrates they make high resolution, real-time imaging of cell–cell interactions possible with minimal interference. In addition, as these devices are modeled to lack the interspecies differences inherent to animal models, they contain the potential to be both complementary and to lessen the dependence on the latter, ultimately reducing costs and ethical complexity [22].

Previous reviews in this journal have focused on the benefits and limitation of vascularized microdevices [23], the use of organoids in microfluidic models [24], and the applications of microfluidics to the study of vascularized tumors [25]. The goal of this review is to describe the advances that have been made in our understanding of blood–endothelium interface biology and pathology as a direct result of vascularized microfluidic models, what challenges and limitations are inherent in our current models for the study of this important interface, and what current advancements in vascularized microfluidic technology might do to impact the future possibilities of these models for the investigation of the blood–endothelium interface. In order to achieve this goal, a brief overview of the fabrication methods involved will be provided first, followed by a detailed discussion of the advances made possible through these models at the plasma, erythrocyte (RBC), platelet, and leukocyte (WBC)–endothelium interfaces. Finally, using these important experiments as a guide, we will conclude by describing the limitations and challenges inherent in current models and what the future may hold for this important experimental platform.

## 2. Fabrication

Perfusable endothelialized microvessels require the creation of perfusable microchannels within a polymer substrate. They can be fabricated using a variety of materials, including solid polymers like polydimethylsiloxane (PDMS) [26] and polyester [27], and can be combined with physiologically relevant hydrogels such as fibrin [28] and collagen [29]. For those laboratories without the expertise or resources necessary to fabricate their own, commercially available models are also available.

For the laboratories fabricating their own devices, one of two general approaches are taken. The first approach is the “bottom-up” approach. In this approach, cells are encouraged to recapitulate the physiologic mechanisms of angiogenesis, the formation of new vessels from pre-existing vessels [30], and/or vasculogenesis, the formation of new vessels with no pre-existing vessels [31]. These pathways are fundamental to in vivo vessel formation [32], where vasculogenesis has been demonstrated to occur through the recruitment of ECs from bone marrow [33], and angiogenesis has been implicated in both physiologic and pathologic responses to stress [34] such as trauma or ischemic events. In these bottom-up fabrication methods, no microstructures are required to guide ECs to form vessels. Instead, ECs are seeded into biologically inspired hydrogels and are coaxed into developing vascular networks.

In the case of perfusable microvascular models utilizing vasculogenesis, ECs can be encapsulated with perivascular cells, such as fibroblasts in hydrogel and exposed to flow. Various stimulating factors including fibroblast-derived proteins [35], such as insulin-like growth factor-binding protein 7, vascular endothelial growth factor (VEGF), and basic fibroblast growth factor (bFGF) [36] have been used to stimulate vessel formation. In addition, co-culturing with other stromal cells has been demonstrated to contribute to vascular stabilization in these models [37], with resultant changes in matrix density able to impact vessel morphogenesis [38]. Various combinations of these stimulatory and stabilizing factors have been demonstrated to influence vessel branch number, branch length, diameter, and vascularized area [39].

Angiogenesis, on the other hand, is a process through which new blood vessels are formed from existing blood vessels, which may or may not include stromal cells. As such, perfusable microvascular models require hybrid approaches, either with vasculogenesis-based bottom-up models or top-down models, which are to be discussed shortly. Angiogenesis is a tightly regulated process in which a series of defined steps are executed: extracellular matrix (ECM) is degraded by matrix metalloproteinases (MMPs), angiogenic stimulating factors are released, vascular sprouting takes place followed by elongation and branching, lumen formation, anastomosis and finally stabilization or regression [40]. While these bottom-up methods have been utilized in modeling the blood–endothelium interface, specifically in the evaluation of physiological flow forces such as interstitial shear stress and intraluminal flow shear stress, they are most commonly used in the study of blood vessel formation and cancer metastasis. The most common draw-back of both bottom-up fabrication approaches is the heterogeneity of vascular size, length, and geometry, which results in uncontrollable flow patterns. While this can limit microvascular experimentation that requires control of vessel size, density, and flow pattern, it better recapitulates the geometry of capillaries and provides unparalleled access to the processes of angio- and vasculogenesis.

The second fabrication approach is the “top-down” approach. In this approach, the pattern of the microvasculature is designed and fabricated before the cells are introduced. Most microfluidic platforms are fabricated in this manner. Within this fabrication philosophy, there are a variety of individual fabrication approaches, the most popular of which is photolithography and soft lithography [41], which are adapted from the semiconductor industry. Photolithography uses photosensitive materials to create desired channel patterns on silicon wafers by shining ultraviolet light through a mask, and soft lithography replicates the patterned channels into microdevices. Other popular top-down methods include 3D printing, in which sacrificial materials such as carbohydrate glass [42] are printed onto a substrate before being removed in a biocompatible fashion, and spatial laser degradation [43], in which a substrate is degraded utilizing a focused, pulsed laser. Because most of these methods require expensive equipment and are time consuming, alternative approaches have been developed that incorporate off-the-shelf materials, such as those which utilize polymethyl methacrylate (PMMA) optical fibers [44] as a mold as illustrated in Figure 1, and the needle extraction method [45]. Both methods cast a polymer such as PDMS around a readily available material in the lab and utilize manual extraction to create channels with rounded lumens, which are typically much more difficult to create using the other fabrication methods. Unlike the bottom-up approach, these fabrication methods allow for tight control of the vascular geometry and size, though the geometries achieved tend to be simplified in order to provide more predictable flow patterns. The most common limitation of this fabrication approach is the difficulty of achieving vascularized channel size on the order of the small microvessels, less than 30 μm in diameter, while also creating complex branching patterns, though this approach can be used to create stenosis, aneurysm, and bifurcation morphologies.

Once the microchannels are created using these top-down approaches they are then endothelialized. In order to accomplish this an ECM protein, such as collagen, fibronectin or laminin is infused and allowed to adsorb to the channels inner surface [46,47,48]. In doing so, the channel is made hospitable for endothelialization. ECs are then introduced into the channels and cultured with EC media under continuous flow conditions resulting in an endothelial monolayer covering the inner surface of the channels.

## 3. Blood–Endothelium Interface

### 3.1. Plasma–Endothelium Interface

The non-cellular component of blood, the plasma, is an aqueous solution of organic molecules, proteins and salts [49]. Not only do the components of plasma carry out constant biochemical interactions with the cellular components of blood, but their interaction with the vascular endothelium is key to the execution of vital functions such as delivery of antibodies in infection, maintenance of tissue pH and osmotic homeostasis. In addition to these biochemical interactions, the plasma executes constant two-way biophysical communication with the vascular endothelium. Within native vessels, ECs are predominantly subjected to three mechanical forces: (1) fluid shear stress, (2) hydrostatic pressure, and (3) cyclic stretch. Indeed, both the cellular–endothelial interactions and the biochemical interactions carried out by the plasma are highly dependent upon the shear conditions at sites of vascular injury or endothelial activation [50,51].

On the biochemical front, one of the more important aspects of the plasma–endothelium interface is the ability of the plasma to deliver drugs and antibodies across the endothelium into tissues. This interface is of particular interest to those studying the very tightly controlled blood–brain barrier (BBB), where both drugs and antibodies have increased impediments to delivery. These impediments include complex tight junctions that restrict paracellular transit [52] and efflux pumps that inhibit the uptake of lipophilic molecules [53]. Using pluripotent stem cell-derived brain microvascular endothelium, astrocytes, and pericytes, Park et al. [54] were able to recapitulate the BBB on a perfusable chip, building upon the work of Campisi et al. [55]. This endothelium expressed high levels of tight and adherens junction proteins and functional efflux pumps with recapitulation of their documented response to Verapamil in vivo, as well as selective transcytosis of peptides and antibodies. Others expanded on the use of perfusable microfluidics to study the BBB, focusing on the role of plasma in electrolyte balance and in particular the use of hyperosmotic therapies for the treatment of central nervous system disease. Linville et al. [56], using stem cell-derived brain microvascular ECs in a PDMS-based model, demonstrated that mannitol, an osmotic diuretic used clinically to treat elevated intracranial pressure, resulted in a dose-dependent, spatially heterogenous increase in paracellular permeability by forming transient focal leaks in the vascular endothelium. In addition, using bFGF before treatment with mannitol, they were able to demonstrate its direct role in the modulation of the degree of BBB opening and subsequent recovery. Such microfluidic models are a great step forward in the study of the BBB and hold immense promise for the live imaging of drug and antibody delivery through an endothelium that is notoriously hard to penetrate.

These EC-cell junctions do not just play a role in vascular homeostasis. They are also integral to the delivery of medications, and in particular can create significant barriers to delivery for large-molecule medications such as biologics. Picking up on the limitation of prior in vitro models, which have tended to show permeability coefficients higher than those seen in vivo [57], Offeddu et al. [58] utilized a bottom-up fabrication approach to develop a self-assembled microvascular network with an average vessel size of 20 μm that recapitulated the morphology and junctional complexity of in vivo microvasculature. They first used 4–500 kDa dextrans as their model for large molecules, as has been done previously, and found that the permeability of the microvessels was two orders of magnitude lower than that described in previous non-vascularized 2D in vitro models, and more in keeping with those that have been seen in in vivo models [59,60]. They then went on to investigate the role of charge on permeability in these large molecules, assessing dextrans of the same molecular weight but different charges, and found that the microvessels had a significantly higher permeability to positively charged molecules when compared to neutrally charged molecules, while the converse was true for negatively charged molecules, in keeping with previous in vivo models [61]. Then, shifting their focus to proteins, using human serum albumin and IgG as model proteins due to their importance as components in biopharmaceuticals, they found that transport was two orders of magnitude lower in microvessels, consistent with previous in vivo models [60,62], and importantly that this behavior was independent of co-culture with fibroblasts, making stromal paracrine contributions less likely. Finally, focusing on the particular pathways these proteins take during transcytosis they turned their attention to the neonatal Fc receptor (FcRn), which is located on ECs and plays a key role in extending the half-life of these proteins in blood [63]. Not only did the microvessels express FcRn, but IgG was strongly co-localized, which is seen in vivo [64]. Using combinations of variation in intraluminal pH as well as small interfering RNA (siRNA) downregulation of FcRn, they found that FcRn did not transport IgG from the lumen to matrix, but instead acted as an efflux mechanism to remove IgG from the intracellular environment, therefore antagonizing IgG transcytosis. Albumin, on the other hand, was not affected by the presence of FcRn. In total, this study provides a strong foundation for the use of microvascular chip models for the preclinical investigation of large-molecule medications such as biologics.

Also integral to the delivery of plasma and its drug and antibody cargo are the ways in which it flows through the microvasculature and the biophysical impact it has on the endothelium. One of the primary ways the endothelium responds to its biophysical environment is the variation in expression of surface proteins and its geometric remodeling in response to these factors, for instance by modulating vascular lumen size through mechanotransduction [65]. This mechanotransduction is mediated through incompletely understood adaptive feedback signaling mechanisms involving mechanosensitive protein machineries on the endothelial surface [66,67]. In the setting of laminar flow, the mechanotransduction triggers the expression of anti-inflammatory, antithrombotic, and antioxidative mediators [68]. On the other hand, when flow is disturbed, there is continuous activation of inflammatory pathways [51], making biomechanical communication between the plasma and vascular endothelium a key focus of investigation for a wide variety of disease conditions. Because of its importance, multiple groups have utilized vascularized microfluidic platforms to explore these processes in real time.

For instance, Mannino et al. [44] demonstrated that variation in the expression of vascular cell adhesion molecule-1 (VCAM-1), a molecule integral to transmigration [69], correlated with wall shear stress dictated by differences in vascular geometry. Others designed intentionally tortuous channel geometries to model turbulent flow and the related increase in shear stress and flow obstruction [70]. In these experiments, the distribution of VE-cadherin, an adherens junction protein [71], became discontinuous in the setting of disturbed flow. In addition, at high shear stress/flow rates both intercellular adhesion molecule-1 (ICAM-1), a major endothelial adhesion protein for WBCs [71], and VCAM-1 expression were attenuated. Not just surface proteins are altered by these vascular geometries and perturbed flow regimes, however. Using a model with a complex three-dimensional microvascular architecture including straight, tortuous, and stenosed geometries, Zheng et al. [72] demonstrated that the secretion of von Willebrand factor and its capacity to assemble complex meshes depended on vascular geometry and fluid flow. Specifically, in vessels less than 300 μm in diameter, with high shear stress, strong flow acceleration, and sharp turns, shown in Figure 2, these von Willebrand factor-induced webs were at their greatest. These webs demonstrated platelet, WBC, RBC binding, and even the shearing of passing RBCs. This model uncovered a set of biophysical parameters that influence microvascular thrombosis—vessel diameter, vessel geometry, fluid shear stress and flow acceleration—and therefore has the potential to serve as an important model for the evaluation of diseases such as thrombotic thrombocytopenic purpura [73].

As mentioned previously, angiogenesis is an important factor in many normal and pathological processes in the body. While it is known that VEGF plays a significant role in this process of vascular sprouting, the complexity of vascular morphogenesis involves an interplay of multiple regulatory factors. Given that the role of plasma flow is so integral to the function of the vascular system, it stands that this same force may play a role in directing angiogenic sprouting, despite biochemical signaling being both necessary and sufficient for angiogenesis. In an effort to investigate this, multiple groups have turned to bottom-up microfluidic models. Using a PDMS-based model of parallel endothelialized channels with intervening collagen gel matrix, for instance, Song et al. [74] found that increased shear stress decreased VEGF-induced morphogenesis and that this attenuation was mediated by nitric oxide (NO). Furthermore, by inducing interstitial flow and coupling that with physiologic levels of intraluminal shear stress (3 dyn/cm^2^), they demonstrated that when ECs invaded the collagen towards the VEGF gradient, but against the interstitial flow, there was an overall increase in the number of filopodia, or tip cell projections, when compared to those cells which invaded away from the VEGF source and toward the interstitial flow gradient. The former morphology is more characteristic of sprouting from an existing vessel while the latter mirrors vessel dilation rather than sprouting. Finally, eliminating VEGF gradient variation the team exposed the ECs to interstitial flow and found that interstitial flow independently enhanced sprouting morphogenesis. Overall, these findings suggest that interstitial flow and shear stress could affect the density and geometry of VEGF-induced sprouting, though this requires further investigation.

Expanding on this focus on intraluminal shear stress, Galie et al. [29] investigated the possibility that plasma flow alone can induce angiogenesis. Focusing on a particular conundrum in the biophysical induction of angiogenesis, there is significant evidence that transmural flow induces the sprouting of ECs [75] while the role of luminal shear in angiogenic sprouting is more controversial, despite both flows exerting shear stress on the endothelium. Using a PDMS-based, manual needle-extraction fabrication method, they found that both transmural and luminal flow induced sprouting at a common threshold of shear stress, namely 10 dyn/cm^2^. They next explored potential underlying mechanisms. Focusing on the role of MMPs, of known importance to cell migration and sprouting [76], quantitative PCR of the sprouting ECs demonstrated that not only was MMP-1 the only member of the MMP family to be up-regulated in the setting of both transmural and luminal flow, but also that its expression increased most dramatically at 10 dyn/cm^2^, while none of the remaining MMPs changed their expression in response to shear stress. Further confirmation of this integral role of MMP-1 was obtained with the confirmation that both a dicer substrate interference RNA (dsiRNA) targeting MMP-1 and introduction of mamaristat, an MMP inhibitor [77], blocked flow-induced sprouting. These experiments demonstrated that both intramural and transmural flow could direct sprouting and that this sprouting took place at a threshold shear stress orchestrated by MMP-1.

While it is understood that this hemodynamic shear stress has impacts on the EC-cell junctions, the mechanism responsible remains poorly understood. Using a combination of vascularized microfluidics and an in vivo murine model in order to leverage the advantages of each, Polacheck et al. [78] attempted to uncover the responsible pathway. With knowledge that shear stress regulates multiple mechanosensitive pathways [15] they focused on the role of Notch signaling, which is required for normal vascular development [79,80,81] and has been shown to be activated by shear in zebrafish [82]. Using a collagen type I solution and an acupuncture-needle retraction method to fabricate microchannels, they identified the Notch1 transmembrane receptor as responsible for regulation of vascular barrier function. After seeding a second model with ECs expressing Notch transcriptional co-factor Mastermind, which inhibits Notch transcriptional signaling, they found that there was no effect on permeability, suggesting a non-canonical regulatory mechanism for Notch in barrier function. Using a combination of CRISPR/Cas-9-mediated knockout of Notch1 and Dll4 (Notch1-KO) in vitro and DAPT, a γ-Secretase inhibitor that cleaves Notch to release the transcriptionally active Notch intracellular domain [83], in vivo they identified that Notch regulates the barrier function of the vascular endothelium through a non-canonical, transcriptionally independent pathway that drives adherens junction assembly. This is done through the formation of a previously unknown receptor complex in the plasma membrane, consisting of VE-cadherin and the Rac1 guanine exchange factor (GEF) Trio, which is known to be involved in Notch-mediated axon guidance in *Drosophila* [84], mediated by the transmembrane tyrosine phosphatase LAR, which is known to bind the Rac1 GEF Trio [85]. Thus, the discovery of a new signaling pathway for vascular barrier function regulation using vascularized microfluidics has opened up an entirely new avenue of research into the study of microvascular barrier function.

### 3.2. Erythrocyte-Endothelium Interface

RBCs are the most abundant cell type in human blood. Through their maturation process they come to lack nuclei, ribosomes, mitochondria and other organelles. In exchange, they accumulate hemoglobin, a protein that allows for the delivery of oxygen to peripheral tissues. Once mature, red blood cells acquire the ability to deform in response to microenvironmental forces [86], allowing them to pass through the narrowest capillaries. This capacity can be severely limited by diseases that alter the mechanical properties of the red blood cell, such as sickle cell disease [87] and spherocytosis [88]. The interaction between RBCs and ECs, furthermore, contributes to the pathogenesis of diseases as varied as sickle cell disease [89] and diabetes [90], making vascularized microfluidics ideal models for their investigation.

At the level of the microvasculature, the interaction of RBCs with the endothelium is its most important. Here, the flow of blood is significantly slowed due in part to the combination of RBC bulk and the endothelial surface layer (ESL), or glycocalyx [91], a gel-like layer of membrane-bound glycoproteins and plasma proteins. The ESL has been noted to be damaged in multiple diseases, including type 1 diabetes [92]. Furthermore, damage to the endothelial glycocalyx can lead to shedding of the red blood cells’ own glycocalyx layer [93], which itself has been implicated in various disease processes such as hemodialysis dependent end-stage renal disease [94]. In an effort to better understand this integral component of RBC-endothelial communication, Tsvirkun et al. [95] developed an endothelialized microvascular model with channels 30 μm in diameter that allowed for measurement of the glycocalyx (600 nm), recapitulating the thickness of its in vivo counterpart, as seen in Figure 3 [96]. Furthermore, they confirmed the existence of a near-wall layer of approximately 4.5 μm away from the apical membrane of the ECs, depleted of RBC and consistent with the cell-free layer (CFL) which has been observed in vivo [97]. This CFL is known to impact both oxygen and NO transport and had not been previously documented in endothelialized microfluidic models, laying the groundwork for exciting future investigations [98].

The role of the ESL in malaria has also been explored using vascularized microfluidics, specifically investigating the role of the endothelial glycocalyx in the adherence of *Plasmodium falciparum* infected red blood cells (PfRBC). After transmission through a bite from an *Anophales* mosquito, malarial parasites pass through a well-orchestrated life cycle. During the blood stage, these parasites reside within RBCs and have their greatest clinical impact on the microcirculation. Not only do these RBCs have altered shape, size, and structure [99], but they alter their membrane properties, resulting in an increased ability to bind ECs [100] as well as other noninfected [101] and infected RBCs [102]. This pathology results in microcirculatory obstruction, leading to anemia and even death. It has been proposed that shedding of the glycocalyx due to local inflammation in the microcirculation may play a key role in malaria’s pathogenesis. By enzymatically removing endothelial sialic acid residues, which are implicated in the initial interactions between PfRBCs and the endothelium, Introini et al. [103] found a twofold increase in the endothelial adhesion of PfRBC. In addition, there was enhanced PfRBC rolling on the diminished endothelium. This experiment supports a proposed role of glycocalyx disruption in malaria and its candidacy as a possible therapeutic target for malaria treatment.

One limitation of these vascularized microfluidic models of malaria, and indeed vascularized microfluidic models in general, is their inability to maintain physiologic barrier function long-term (weeks to months) so that both endothelial dysfunction and restoration of barrier function can be investigated. In an effort to address this shortcoming and apply it to the investigation of both malaria and sickle cell disease (SCD), Qiu et al. [104] employed an agarose-gelatin interpenetrating polymer network (IPN), which confers the advantages of tight control of microvascular size and flow dynamics as well as long-term culturing with physiologically relevant endothelial permeability [105]. Using standard photolithography and modified soft lithography, microvessels of as small as 20 μm were fabricated and, unlike in PDMS counterparts, the ECs cultured using IPN assembled a physiologic basement membrane. Perfusion of PfRBCs alone resulted in a two-fold increase in microvascular permeability to albumin at the locations of PfRBC cytoadhesion, and this increased permeability was reversible, recovering to baseline at two days. In a separate experiment, the synergistic impact of humoral and cellular elements on endothelial barrier function were investigated. Using tumor necrosis factor alpha (TNF-α) to activate the endothelium, PfRBCs were perfused resulting in enhanced microvascular occlusion. Unlike the non-activated endothelium, barrier function did not return. This same model illustrated a similar impact of SCD RBCs. Finally, focusing on a byproduct that impacts both malaria and SCD, extracellular heme, a hemolytic byproduct, when perfused in isolation without the confounding in vivo elements of cytokines or activated cells, resulted in increased EC permeability in a dose-dependent manner, recapitulating in vivo models [106].

The reason SCD RBCs cause such damage in the microvasculature through vaso-occlusion and hemolytic anemia is the propensity of sickle (HbS) hemoglobin to polymerize under low oxygen tension [107]. This process changes the shape and rigidity of the RBC membrane as well as its adhesiveness [108]. During the complex cascade of events involved in vaso-occlusion and hemolysis a variety of membrane-bound proteins on both the endothelial and RBC surface are upregulated in response to the inflammatory cascade. One of these, very late antigen-4 (VLA-4) helps orchestrate blood cell-blood cell and blood cell-endothelial adhesion, and is expressed at higher levels in SCD patients with frequent vaso-occlusive crises and is decreased in patients treated with hydroxyurea (HU) [109,110], a drug used for both the treatment and prevention of vaso-occlusive crises [111]. With this in mind, White et al. [112] utilized a commercially available microfluidic platform to model SCD and investigated the therapeutic potential of VLA-4 in the treatment of SCD. Using whole blood and endothelium activated with TNF-α the VLA-4 blocking antibody natalizumab achieved dose-dependent inhibition of over 50% of whole blood cell binding to activated ECs under physiologic shear flow. These findings support the further investigation of VLA-4 blockade as a novel therapy for SCD.

Others have focused on the biophysical impact of standard therapies on SCD vaso-occlusion. Tsai et al. [26], using human lung microvascular ECs (HLMVECs), as pulmonary postcapillary venules are a common site of vaso-occlusion [113], demonstrated the conferred advantage of HU on sickle blood velocity when compared to untreated SCD patients. HU is effective in the prevention and treatment of vaso-occlusive crises by decreasing the polymerization rate of HbS [114], which in turn decreases adhesion [115] and the number of non-RBC cells that promote vaso-occlusion [113], and promotes NO which leads to vasodilation [116]. Unfortunately, HU also increases both hematocrit and RBC cellular volume [117], which raises concern for its contribution to increased blood viscosity. Using the same model, blood from patients taking HU was found to result in a significantly lower number of obstructed microchannels when compared to treatment naive patients, suggesting that treatment with HU offsets the theoretical risk from increased viscosity as a result of higher hematocrit.

Though HU is known to alter adhesion, it is not the only factor responsible for alterations in EC expression of adhesion molecules. Physical factors, such as disturbed flow or shear stress gradient resulting from altered vascular geometry, as well as intravenous fluid tonicity, also play a role. In an effort to investigate the role of altered vascular geometries in adhesion and vaso-occlusion in SCD, Mannino et al. [44] demonstrated that spatial variation in vascular cell adhesion molecule (VCAM-1) expression correlated directly with disturbed wall shear stress, and in addition that RBC in SCD patients exhibited increased adhesion behavior at the high wall shear stress of bifurcations. In a separate experiment, tonicity of perfused intravascular fluid was altered to mimic the intravenous fluids given to SCD patients in crisis and was found to alter not only adhesivity, but also deformability and their propensity to sickle [118]. Specifically, fluids with higher tonicity (sodium = 141 mEq/L) decreased sickle red blood cell deformability, increasing occlusion and adhesion. Hypotonic fluids (sodium = 103 mEq/L), on the other hand, decreased sickle red blood cell adhesion, but prolonged transit time due to swelling. Intermediate tonicity fluids (sodium = 111–122 mEq/L) resulted in optimization of sickle red blood cell biomechanics, reducing the risk for vaso-occlusion.

In addition to intravenous fluids, patients with SCD routinely require transfusions with allogenic RBC. These can be lifesaving when hematocrit levels are critical. Unfortunately, transfusions have been associated with increased mortality and morbidity in the general population of critically ill patients [119], and the pulmonary microvasculature is hypothesized to be the culprit [120]. In order to investigate the role of RBC-induced damage in the pulmonary microvasculature that might result from transfusion, Seo et al. [121] fabricated a vascularized microfluidic model that incorporated a mechanical stretching system to mimic respiration. This model demonstrated that the perfusion of primary human pulmonary microvascular EC lined channels with RBC resulted in the abnormal cytoskeletal rearrangement of RBC and release of the EC intracellular protein high mobility group box 1 (HMGB1), a DNA-binding protein released during necroptosis of ECs that then acts as an inflammatory cytokine [122] and has been implicated in vascular injury in the setting of transfusion [120]. Furthermore, by altering the shear stress at which RBC were perfused they found that lower hemodynamic shear stress (0.14 dyn/cm^2^) resulted in significant loss of intracellular HMGB1, suggesting the low microcriculatory flow seen in critically ill patients may predispose to vascular injury. Finally, by incorporating the mechanical stretching experienced by the pulmonary microvasculature during respiration, they demonstrated that the physiologic deformation of the endothelialized channel significantly increased extracellular release of HMGB1 in the setting of RBC transfusion, but not when being perfused with culture medium. This suggests that ventilation may play a role in transfusion-induced microvascular injury in the lung.

### 3.3. Platelet–Endothelium Interface

Platelets are, like RBC, blood cells without nuclei. They are derived from fragments of bone marrow megakaryocytes and circulate preferentially at the periphery of the blood stream due to mass effect from the much larger RBC. In this advantageous position they serve their primary homeostatic function - surveillance of the vascular endothelial barrier [123]. When a breakdown in this barrier is encountered, platelets are exposed to subendothelial proteins [124], such as von Willebrand factor (vWF) and collagen, which cause them to adhere to the vessel wall, aggregate and crosslink with fibrin to form an occlusive plug. While the intact vascular endothelium expresses molecules that inhibit platelet adhesion, such as prostacyclin and NO, and prevent coagulation, such as thrombomodulin, an injured vascular endothelium releases vWF multimers from Weibel-Palade body granules [125,126], which recruits platelets from the blood stream to form and expand clot in a shear-dependent manner [50].

While thrombosis and hemostasis is the primary role of the platelet, it is also integral to the inflammatory response [127], maintenance of vascular integrity [123], and angiogenesis [128,129], as well as being implicated in multiple diseases, including sickle cell disease [130], hemolytic uremic syndrome [131] and cardiovascular disease [132]. The function of platelets in both homeostasis and these pathologies is profoundly altered by blood flow dynamics [133], making the use of perfusable microvascular models an ideal in vitro platform for the evaluation of platelet-endothelial interactions and their role in human health and disease.

As the primary role of platelets, thrombosis and hemostasis has been the dominant focus of perfusable microfluidic model translation. In an effort to make a clinically relevant model of bleeding to study this process, Sakurai et al. [134] developed the first microfluidic bleeding model that recapitulates key aspects of in vivo mechanical injury at the microvascular level. Using a PDMS-based model, a pneumatic valve was introduced that induces vascular injury when engaged. By altering shear conditions during induced vascular injury they found that bleeding time was dependent upon vWF, known to be a shear-responsive protein [135], at high shear only, and that this behavior was mediated through the limitation of platelet accumulation. Furthermore, by incorporating the anti-platelet agent eptifibatide, primarily used in the setting of acute cardiac ischemia [136], they demonstrated that while eptifibatide did not impact in vitro bleeding time, it did dramatically alter the cellular architecture of human hemostatic plug formation, decreasing clot contraction and therefore lowering the density of platelets within the hemostatic plug. Vascularized microfluidics have also been adopted for the investigation of non-mechanical forms of microvascular injury. For instance, Sylman et al. [137] used a surface microelectrode as a heat source for the induction of heat injury. The tight control of the surface electrode allowed for a spatiotemporally defined zone of prothrombotic (thermally injured) and antithrombotic (uninjured) ECs. This model confirmed that heat activated ECs induce platelet accumulation through the secretion of vWF, providing a new model for the in vivo investigation of a clinically relevant mechanism for microvascular thrombosis in burn patients [138].

The process of endothelium-induced vWF secretion is not the only important thrombogenic variable in the setting of injury. The hemodynamic environment of the endothelium itself has significant bearing on thrombogenesis, where shear gradient can drive platelet-mediated thrombus formation [139]. Multiple vascularized microfluidic models have been developed to further elucidate this relationship. Using a series of parallel microchannels with varying luminal stenosis in a PDMS mold, Westein et al. [140] demonstrated that post-stenotic ECs markedly upregulate surface expression of vWF, and additionally that this expression resulted in a 15-fold increase in vWF-multimer surface area, which could not be achieved in non-stenotic models regardless the level of induced shear. When whole blood was perfused through these stenotic channels substantial platelet aggregation resulted along the post-stenotic, vWF-upregulated endothelium. In a later study, Mannino et al. [44] expanded on this mechanism. Further, using a PDMS-based model of stenosis, the question of what role the endothelium itself plays in post-stenotic platelet aggregation was investigated. First, using a bare channel, the preference of platelets to accumulate in the post-stenotic, high-shear portion of the lumen was reconfirmed. The same model was then coated with a HUVEC monolayer. Perfusion of platelets through this newly endothelialized channel resulted in an overall significant decrease in platelet aggregation throughout the microvessel, regardless the local shear environment.

Shear also plays an important role in platelet-aggregation in various hematologic diseases. In the case of SCD, considered mostly the purview of the RBC, vascularized microfluidics have clarified the important role of platelets. Using a confluently endothelialized multi-shear microfluidic device the whole blood of patients with SCD as well as normal controls was perfused at varying shear rates [141]. Not only did platelets aggregate in a non-shear dependent fashion at a higher flow rates in SCD patients, but this pro-thrombotic behavior was attenuated in patients on HU therapy. This same impact was seen in the case of hemolytic uremic syndrome (HUS), a thrombotic microangiopathy classically associated with enterocolitis from Shiga-toxin producing *Escherichia coli* (STEC). Tsai et al. [26] developed a vascularized microfluidic model and infused Shiga Toxin 2 (STX2), a toxin that induces the hematologic manifestations of HUS [142]. They observed that the resulting thrombi formed consisted of WBCs, platelets and vWF. Furthermore, these thrombi were larger and caused microchannel occlusion faster at higher shear rates, suggesting that the microvascular thrombosis seen in HUS is shear dependent. Finally, the infusion of the platelet-inhibitor eptifibatide resulted in significant abatement of thrombus formation and microchannel occlusion. This effect, like thrombus formation, was most pronounced at high shear rates.

In addition to its role as a physiologic response to bleeding and a pathologic byproduct of disease, thrombosis can also occur as a complication of therapeutics. Given the complexity of in vivo models, uncovering the mechanisms by which these events occur can be difficult. In the case of Hu5c8, a monoclonal antibody against CD40L intended for the treatment of autoimmune disorders [143], vascularized microfluidics were used to address this shortcoming. In clinical trials, Hu5c8 caused unexpected, life-threatening thrombotic complications that were not discovered during preclinical investigations [144]. They hypothesized that this adverse effect was due to the Fc domain of the medication, which preferentially bound the FcγRIIa receptor, a receptor specifically expressed on platelets and implicated in fibrin clot formation [145]. Using a whole-blood perfused vascularized microfluidic model, they infused the Fc-domain altered antibody and uncovered that the thrombotic complication seen in the original molecule was not present. This model not only demonstrated the value of vascularized microfluidics for recapitulation of key aspects of the coagulation cascade, but also its strengths as a preclinical model for drug discovery.

### 3.4. Leukocyte–Endothelium Interface

Leukocytes, unlike RBCs and platelets, are nucleated blood cells. While their primary role is as mediator of the immune and inflammatory responses, they take on various identities with unique structures and functions [6]. In the vascular system, WBCs encounter the endothelium as a simple result of flow dynamics [146], much like platelets. At the endothelial interface, in response to endogenous or exogenous stimuli, WBCs follow a well-choreographed series of margination steps including rolling, which is mediated by selectins [147], and activation and adhesion, which are both mediated by integrins [148,149]. Once adhesion takes place, transendothelial migration can occur either paracellularly or transcellularly. It is important to note that WBCs are not alone in this communication with the endothelium. Platelets, discussed above, are integral to both inflammation and immunity [150]. Not only do they increase endothelial activation [151], they also release potent substances that stimulate WBCs [132] and promote both adhesion and transmigration [152]. The primary site for WBC recruitment and WBC-platelet interaction takes place within the postcapillary venules [153], a scale at which endothelialized microfluidic devices are well suited as a model for investigation.

The physical properties of WBCs are of particular importance to their function in the circulation. In the case of their flow through vascular beds, for instance, it has been proposed that WBC stiffness contributes significantly to margination behavior [154]. Common medications such as glucocorticoids as well as endogenous catecholamine hormones are known to induce demargination of WBCs and increase white blood cell count, as well, and though the underlying mechanism is not well understood, previous experimental and computational modeling of margination has suggested that membrane stiffness plays a role in driving WBCs toward the vessel wall [155,156]. Using two different microfluidic models, a non-endothelialized model with a vessel diameter of 5.9 μm (+/− 0.08 μm), smaller than the typical WBC size, and an endothelialized model measuring approximately 150 μm in diameter, Fay et al. [157] expanded on this understanding through the discovery of two important aspects of WBC demargination behavior. Firstly, after administering glucocorticoids, the proportion of granulocytes that traversed and demarginated from the microfluidic model of capillary beds and veins, respectively, increased. Secondly, both glucocorticoids and catecholamine exposure reorganized the cellular cortical actin, reducing granulocyte stiffness, and using a kinetic theory computational model, they demonstrated that this reduction in stiffness is alone sufficient to cause demargination. This has important implications for our understanding of the inflammatory cascade as well as both hemopoietic stem cell mobilization and homing. The geometry of the vascular endothelium itself has additional important implications for the choreography of WBC margination, which was investigated using vascularized microfluidics by Khan et al. [70]. To explore this interaction between vessel geometry and WBC activation, they fabricated a platform with built-in branching and flow obstruction. Then, using a THP-1 WBC model and monoclonal antibodies to E-selectin and ICAM-1, expressed on ECs, they found that areas where multiple channels converge were the most prone to WBC attachment.

Expanding on the capabilities of microfluidic models others took to modeling the efficacy of anti-inflammatory therapeutics on atherosclerotic plaque formation. Using inhibitors of TNF-α and the nuclear factor kappa-light-chain-enhancer of activated B cells (NF-kB) signaling pathway, Chen et al. [158] demonstrated that, when compared to endothelium not activated with TNF-α, WBCs had > 74% less adhesions and > 87% less transendothelial migration events. Others expanded on these findings by using TNF-α to activate endothelium in a novel pneumatic-controlled stenosis model, recapitulating the role of inflammation in the formation of atherosclerotic plaque [159]. Using THP-1 WBCs, it was demonstrated that endothelium in the stenotic regions expressed higher levels of ICAM-1, a key WBC adhesion marker, compared to other regions. In addition, in both activated and non-activated endothelium increased WBC rolling and adherence was noted in the stenotic regions, highlighting the importance of endothelial activation, flow disturbance and vessel geometry to the atherogenic microenvironment.

Leveraged microfluidic models have also been used to investigate the role of angiogenesis in atherosclersosis. In the setting of atherosclerotic plaque maturation neovascularization takes place. This primarily originates from adventitial vasa vasorum which invade the intima via sites of media disruption [160] in response to stimuli originating from the plaque itself. This process of angiogenesis is an important source of intraplaque hemorrhage [161], though is not well understood. There is, however, increasing evidence that inflammation contributes significantly to the pathogenesis of a variety of angiogenic diseases [162], including atherosclersosis [163]. Because of the difficulty with quantitative analysis and tight control of both vascular geometry and diffusion parameters in previous models, groups have turned to hybrid top-down/bottom-up endothelialized microfluidic platforms for investigating this seemingly important role of WBCs in angiogenesis.

Focusing specifically on the role of neutrophils in angiogenesis, Wu et al. [164] investigated the bidirectional communication between vascular ECs and neutrophils and the impact of this communication on cellular migration. First, using gradients of IL-8, a neutrophil chemokine found in a number of angiogenesis-associated diseases [165], and VEGF, they demonstrated that endothelial migration is initiated by VEGF, and that this migration increases the number of neutrophils that are able to migrate along the chemokine gradient. Then, by introducing a drug that inhibits endothelial migration, endostatin [166], they found that despite the expected decrease in projected EC migration area in experiments without neutrophils, when neutrophils were added there was no significant decrease in neutrophils within the endothelial migration channel. Expanding on this finding of neutrophil-mediated stabilization of the EC structure during angiogenesis, they then varied IL-8 and VEGF gradients and found that the neutrophil-mediated reduction in endostatin impact on EC migration is regulated by chemokine concentration and not VEGF. These findings suggest endostatin may have a therapeutic role in inhibiting angiogenesis in highly angiogenic disorders such as cancer.

The role of WBCs in the formation of blood vessels is not limited to healthy cells. While the study of angiogenesis in the setting of malignancy has been overwhelmingly focused on the process of solid tumor growth and metastases, hematologic malignancies such as leukemia are also known to induce angiogenesis, particularly within the bone marrow [167]. The pathophysiology of this process, however, is only just beginning to be investigated, up until this point predominantly using 2D cultures [168]. In an effort to increase the generalizability of these preclinical investigations, Zheng et al. created a PDMS based microfluidic device using soft lithography with type 1 collagen gel as the ECM through which angiogenic sprouting would take place [169]. The model consisted of two channels separated by collagen ECM, one seeded with ECs and the other with leukemic cells, as illustrated in Figure 4. They used three different leukemic cell lines derived from patients with histiocytic lymphoma (U937), acute promyelocytic leukemia (HL60), and chronic myelogenous leukemia (K562), all of which are known to secrete VEGF and other proangiogenic factors. The absence of leukemic cells resulted in no vascular sprouting by ECs due to the lack of proangiogenic factors. Upon introduction of the leukemic cells, however, angiogenic sprouting occurred in all tumor cell types, though the morphology of each was distinct. In particular, the chronic myelogenous leukemia cell line resulted in the deepest invasion and highest number of tip cells, ECs that lead the tips of vascular sprouts.

Leukemic cells are not alone in the bone marrow, however. Various stromal cells, including bone marrow fibroblasts, provide structure for the leukemic cells and are known to secrete growth factors, chemokines and ECM to mediate the bone marrow microenvironment and even induce angiogenic sprouting [170,171]. In order to evaluate the impact bone marrow fibroblasts have on leukemia-mediated angiogenesis the bone marrow stromal cell line HS5 was introduced into the leukemic channel without the leukemic cells present. By day 3, the stromal cells alone induced significant angiogenic sprouting, and unlike the leukemic cells, these sprouts were multicellular with mature lumens rather than single, isolated ECs, in keeping with what is seen in biopsies of healthy bone marrow [172]. When co-cultured with HL60 or K562 cells, greater endothelial invasion distance and area were seen than either in isolation, but with more isolated tip cells when compared to the bone marrow fibroblasts alone, consistent with bone marrow biopsies taken from patients with acute myeloid leukemia [173].

## 4. Limitations, Challenges & Future of Vascularized Microfluidics

The advent of vascularized microfluidics has brought about a rapid expansion in our understanding of the ways in which blood cells and the vascular endothelium interact to protect against, enable, and cause a variety of disease processes. By virtue of their ability to perfuse blood under physiologic shear in optically transparent microchannels approaching the size of in vivo capillaries, they have for the first time made possible the real-time visualization of blood–endothelial cell interactions in a physiologically relevant environment. These models, however, are limited by their reductionism, which is simultaneously their primary advantage. Even as fabrication and biological complexity of these models increases, they are not designed to accomplish complete recapitulation of biological complexity. As a result, the advances made using these models require parallel in vivo experiments to ensure reproducibility and also to gain traction as a biologically legitimate platform to use for preclinical investigation moving forward. In addition to this limitation, there are many challenges, each of which prevents the recapitulation of an important aspect of the in vivo microvascular environment. These challenges span the biological, chemical, mechanical, and usually some combination thereof, and are impacted by fabrication, some features of which are illustrated in Table 1 and further expanded upon in the following paragraphs.

### 4.1. Endothelial Cell Diversity

The ECs being cultured have significant implications for the generalizability of findings during vascularized microfluidic experimentation. HUVECs have been used in the preponderance of models because of their well-studied gene expression profiles and physiology. ECs are heterogeneous throughout the body, however, both in physiology and functionality [1]. Furthermore, it is not well understood whether the use of mature organotypic ECs or pluripotent stem cells are more suitable to these models, and these variations are only just beginning to be explored. Though some models discussed heretofore have taken advantage of organotypic ECs [121] and pluripotent stem cells [54], tapping the potential of vascularized microfluidics is dependent upon their expanded use.

One type of ECs in particular has been noticeably absent from a majority of vascularized microfluidics, and yet it is a vitally important component of the microvascular system—lymphatic ECs. As described in multiple models above, transmural flow of intravascular fluid is a critically important function of the endothelium in both homeostatic and pathologic settings. The primary mechanism by which the body removes this interstitial fluid is through the lymphatic system [174]. In addition, it serves to traffic immune cells and absorb lipids, among other functions. Despite these important roles, lymphatics have been mostly ignored in the fabrication of vascularized microfluidics, though recently groups have been making advances in modeling them in isolation on functionalized chips [175]. The goal of vascularized microfluidics to recapitulate a truly physiologically and functionally relevant microenvironment will require the eventual incorporation of lymphatics.

### 4.2. Perfused Biofluids

Of equal importance in the generalizability of vascularized microfluidic experiments to the culturing of diverse ECs is the perfused blood substitute, or biofluid. These biofluids span the completely artificial, as in the case of culture media [54], to the near recapitulation of in vivo systems, as in the case of whole blood [134], with the majority of models employing an intermediate biofluid in the form of cell suspensions [121]. The choice of biofluid reflects the advantages of reductionism in vascularized microfluidics toward the investigation of highly specific blood–endothelial interactions. This reductionism comes at a cost, however, and while some models have taken full advantage of whole blood in an effort to recapitulate the physiology of in vivo microvasculature, the inherent adhesion and activation of blood cells as well as their contact with the hardware required for perfusion necessitates anticoagulation. Anticoagulation, however, has inherent limitations, particularly in the study of thrombosis and hemostasis where platelet aggregation and activation as well as initiation of the coagulation cascade can be either abnormal or absent. Expanding the use of whole blood and engineering ways to adapt its limitations, nonetheless, will be an important component of ensuring the generalizability of findings generated by preclinical investigation using vascularized microfluidic models moving forward.

### 4.3. Microvascular Size, Geometry and Dimensionality

Modeling vessels of varying sizes and geometries is also of upmost importance to the recapitulation of not only the in vivo rheology of blood but also the biophysical environment encountered by blood cells at their site of greatest communication with peripheral tissues such as stenoses and bifurcations. The microvasculature consists of arterioles, capillaries and venules which have on average a luminal diameter ranging from approximately 300 μm to the width of a single RBC. The currently available fabrication methods utilized in the development of vascularized microfluidic models, however, have limitations with regards to both channel size, particularly at the smallest diameters, and geometry. Perfusable channel sizes range from tens of micrometers in the case of photolithography-based models using PDMS as a substrate [46] and “bottom-up” approaches [58] to on the order of 200–400 μm in the case of inkjet [176] and 3D printing fabrication methods [177]. The ability to create microchannels at both ends of the size spectrum come with limitations, however.

Firstly, developing an intact EC monolayer in the smallest channels can be unreliable due to channel clogging. In addition, the fabrication methods used to develop these small channels are either unable to allow for geometric complexity at this scale, as in the case of photolithography, or provide little to no control over vessel geometry and flow regimes, as in the case of “bottom-up” approaches. On the other end of the spectrum, fabrication methods employed for the largest microvessels and smallest macrovessels allow for modelling of complex geometries, but have limitations related to EC seeding in a setting where gravity has a significant impact on cell distribution and can hinder monolayer formation by inducing EC pooling at the bottom of channels, though strategies such as rotational seeding have been employed to combat this challenge [44].

The choice of substrate material, in addition, is paramount. The mechanical properties of the substrate and the geometries created through fabrication have both been demonstrated to alter surface protein expression and permeability of the endothelium [44]. Though advances in achievable geometric complexity are being made using novel approaches such as photopolymerizable hydrogels [178], the ability to manufacture geometric complexity while also recapitulating capillary size remains elusive.

Finally, many endothelialized models’ attempt at recapitulation of the blood–endothelium interface, spanning investigations into numerous organ systems including the lung [179], liver [180], and brain [181], have been limited to two dimensions. As a result, these models have inherent limitations in generalizability from a geometric perspective. For instance, these models lack the ability to recapitulate the normal curvature and size scale of the microvasculature and in addition suffer from abnormal activation of coagulation and blood cells as a result of exposure of the non-endothelialized surface, both altering the relationship of the blood to the endothelium in ways that can confound conclusions. Such models have allowed for reductionist recapitulation where no alternatives were possible and now serve as steppingstones to more biologically accurate designs.

### 4.4. Temporal Integrity

In addition to the challenges of size and geometry, a majority of the vascularized microfluidic models discussed heretofore face the additional hurdle of temporal integrity, which is predominantly dependent upon substrate choice. This challenge has the impact of limiting models to the study of acute processes that do not need to take into account long term vascular integrity and remodeling. Much of this challenge is a consequence of the reliance on PDMS as a model substrate. Its stiffer mechanical properties compared to native blood vessel tissues and propensity to absorb and swell when exposed to hydrophobic molecules are significant hurdles [182]. As a result, PDMS is unable to sustain long-term culture without using additional ECM substitutes such as IPN hydrogels [104,183], which allow for control of ligand density as well as ECM elasticity and porosity [184]. Using these novel ECM substitutes models have achieved temporal integrity of up to eight weeks under physiologic flow [104], though much work must be done to achieve culture times amenable to the investigation of the most chronic of conditions.

### 4.5. Recapitulation of Physiologic Microenvironment

The microvasculature consists of more than just blood, endothelium and ECM, however. As emphasized in multiple experiments outlined heretofore, stromal cells such as fibroblasts and pericytes are integral to a healthy and functioning vessel and can through their paracrine functioning alone have a significant impact on microvascular homeostasis. Despite this, they have not been an integral part of most vascularized microfluidic models. This is because the isolation and culture of these cells pose particular challenges. For instance, pericytes share many biological markers with other cells, making their isolation difficult [185]. Furthermore, their phenotype is highly dependent upon culture conditions, and the resultant phenotypic variation can hinder fabrication uniformity [186].

Even when stromal cells are able to be cultured reproducibly, their integration into microfluidic models pose challenges. During fabrication, it must be ensured that stromal cells are positioned on the periphery or abluminal side of the ECs so as not to be directly exposed to fluid flow. Barriers to accurate delivery and incorporation include currently employed ECM substitutes, which limit the ability to encapsulate multiple cell types into a vessel wall. In addition, these materials tend to alter the propensity of stromal cells to synthesize and lay down the ECM so important to their function in vivo. Because of the role of the ECM in mechanical integrity, for instance, poor recapitulation of its in vivo properties by not including cell-derived ECM has implications for the ability to generalize findings. Advances, however, are being made in the incorporation of non-ECs into vascularized microfluidics, not only pericytes and fibroblasts, but also vascular smooth muscle cells [187]. Through the use of hydrogel materials, such as collagen and methacrylated gelatin, vascular smooth muscle cells can be encapsulated [188].

Advances in novel hydrogel materials, however, must be made in both their compatibility with current microfabrication techniques, where when combined with PDMS novel hydrogels should prevent detachment from PDMS during experimentation, and their sustained mechanical properties while allowing for cellular remodeling at the same time [105]. Some have stopped using PDMS-based hybrid models altogether instead opting for the use of only collagen [29], fibrin [189], agarose [190] or alginate [191], which allow the tuning of stiffness and permeability. No ideal hydrogel exists, however. Some suffer from failure in the ability to decouple stiffness and ligand density, while the others are not degradable or suffer from uncontrolled swelling [105].

## 5. Conclusions

The future of vascularized microfluidics is bright, as evidenced by the contribution these platforms have already made to our understanding of the biology of the blood–endothelium interface. The challenges are not insurmountable, and the potential for truly groundbreaking discovery is immense. The path forward will be driven by two complementary approaches: addressing the challenges and limitations of current materials and using a form-follows-function approach to investigate questions posed by the latest scientific advances in microvascular biology. In doing so, not only will meaningful advances be made in the understanding of the microvascular environment, but a robust platform will be reinforced and be incorporated into the standard arsenal of complementary preclinical tools over time.

## Figures and Tables

**Figure 1 micromachines-11-00018-f001:**
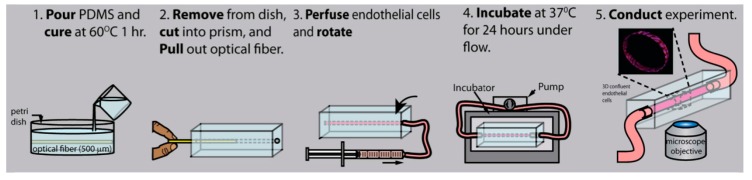
Fabrication steps using polymethyl methacrylate (PMMA) optical fibers in “top-down” fabrication. Adapted with permission from [44].

**Figure 2 micromachines-11-00018-f002:**
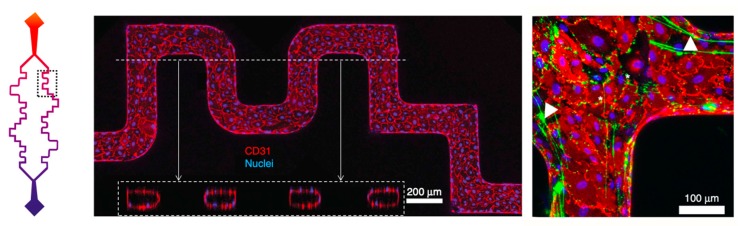
Schematic of pre-designed microvascular geometry (left) with confocal z-projection images (center) demonstrating CD31 (red) and nuclei (blue) staining in the microchannel wall. This geometry induces significant shear stress on the endothelium making them useful for evaluating its impact on vWF mesh formation (right) indicated with arrows and stained green. Adapted with permission from Zheng et al. [72].

**Figure 3 micromachines-11-00018-f003:**
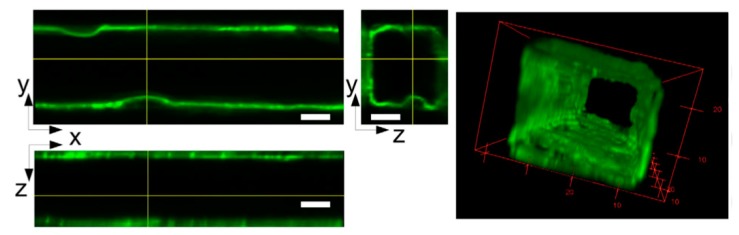
Confocal images depicting the wheat germ agglutinin-tagged ESL in three planes (left) with a 10 μm scale bar. On the right is a 3D rendering of the ESL. Adapted with permission from Tsvirkun et al. [95].

**Figure 4 micromachines-11-00018-f004:**
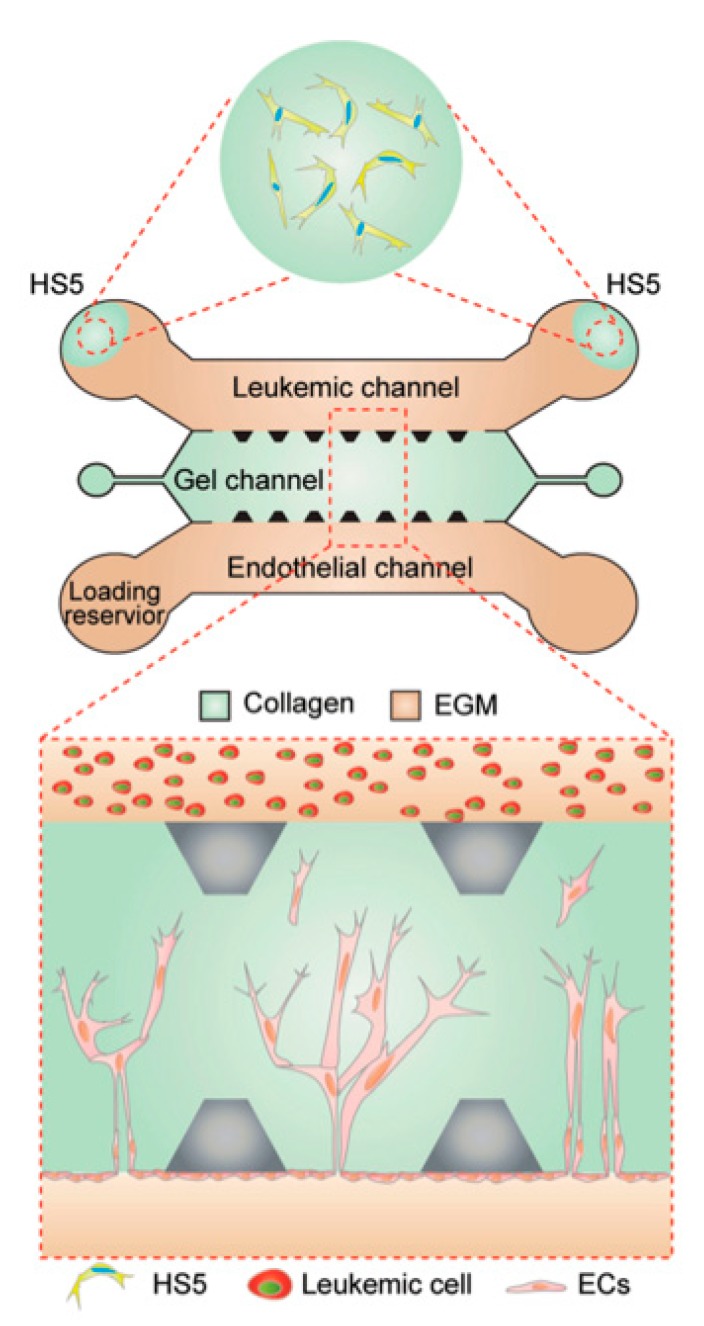
Schematic of an angiogenic microfluidic platform with three parallel channels, the central channel of which contains collagen gel. This acts as a barrier between the ECs and leukemic cell channels. In assays using stromal cells, cells from the bone marrow stomal cell line HS5 were injected into the leukemic cell channel prior to loading leukemic cells. The zoomed in illustration depicts the endothelial to leukemic directionality of angiogenesis seen in the model. EGM = EC Growth Medium. Adapted with permission from Zheng et al. [169].

**Table 1 micromachines-11-00018-t001:** Vascularized microfluidic devices outlined in this review and their ability to incorporate various factors important to microvascular blood cell–endothelium interface experimentation. ✓ = present; o = not present; ? = unclear; * 5–10 μm; ** Not a straight channel.

Author, Year [Ref.]	Non-HUVEC EC	Stromal Cells	Whole Blood Perfusate	Capillary-Sized Microchannels *	Complex Geometry **
Plasma					
Song, 2011 [74]	o	o	o	✓	✓
Galie, 2014 [29]	o	o	o	?	✓
Zheng, 2015 [72]	o	o	✓	o	✓
Polacheck, 2017 [78]	✓	o	o	o	o
Park, 2019 [54]	✓	✓	o	?	✓
Linville, 2019 [56]	✓	o	o	o	o
Offeddu, 2019 [58]	o	✓	o	✓	✓
RBC					
White, 2016 [112]	o	o	✓	o	o
Tsvirkun, 2017 [95]	o	o	✓	o	o
Carden, 2017 [118]	o	o	✓	✓	✓
Seo, 2017 [121]	✓	o	o	o	o
Introini, 2018 [103]	o	o	o	o	o
Qiu, 2018 [104]	✓	o	o	o	✓
Platelet					
Westein, 2013 [140]	o	o	✓	o	✓
Sylman, 2015 [137]	o	o	✓	o	o
Sakurai, 2018 [134]	✓	o	✓	o	o
Barrile, 2018 [144]	o	o	✓	o	o
Leukocyte					
Fay, 2016 [157]	o	o	✓	o	✓
Zheng, 2016 [169]	o	✓	o	?	✓
Wu, 2017 [164]	o	o	o	?	✓
Chen, 2018 [158]	o	o	✓	o	✓
Multiple					
Khan, 2011 [70]	o	o	o	o	✓
Tsai, 2012 [26]	✓	o	✓	o	o
Mannino, 2015 [44]	✓	o	✓	o	✓

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
