# Peer review of "Vascularized Microfluidics and the Blood–Endothelium Interface"

_micromachines, 2019, doi:10.3390/mi11010018_

Round 1

Reviewer 1 Report

General

This review provides useful and interesting perspectives on how in vitro systems have provided new insights into the interaction between blood and its constituents, and the endothelial lining.  This represents a somewhat different approach from other reviews of in vitro vascularization studies, and is therefore a useful addition to the literature.  

Suggestions for improvement:

Past and present tense have been used interchangeably to discuss published research.  Past tense is preferred. 

Page 2.

paragraph beginning line 49. This paragraph does not give enough credit to a history of scientific work using methods other than vascularized microfluidics.  The authors dismiss the contribution of in vivo and “conventional” in vitro work without respecting the body of knowledge that has come from these works.  It would be worth briefly discussing the value of in vivo experiments before enumerating the benefits of in vitro. Line 55-56 “… its phenotype is not that of human endothelium. This has led to conflicting results. ”: In what way?  Elaborate with support from the literature. Line 59 “relatively cursed”: This phrase is too informal. Line 59-60. There have been in vitro models incorporating fluid flow before the development of microfluidic models.  See parallel-plate chambers and cone-plate rotational viscometer systems from the 80s.  Line 66-67: Line 60 “lessen the dependence…” Do they really? Animal studies are often still considered the gold standard and in vitro studies are not free of animal-derived products.

Page 3.

The descriptions of vasculogenesis and angiogenesis are poor and only relate to the in vitro occurrence of these processes. The manuscript would be improved by rewriting to explain the biology of vasculogenesis and angiogenesis, then describing how it is recapitulated in vitro to create vascularized models.  “top-down” vs “bottom-up” terms would be better as descriptive terms such as “patterned” vs “self-assembled”. The “do-it-yourself” method name is too colloquial and misleading since all labs that are making vascularized devices are doing it “in house” unless purchasing commercial ones.  There was no mention of commercially available alternatives (such as those by Ibidi or Aim Biotech) for labs without the expertise to fabricate their own. Authors appear biased toward their preferred method. Please discuss strengths and limitations of all methods more objectively. The benefit of channel molding seems to be simple fabrication of uniform, round vessels that can be seeded with an endothelial monolayer. This is advantageous for creating predictable flow pattern, but no more biologically faithful than other approaches which focus on other feature of vascular biology. Final sentence. Make it clearer that this method can only be used to create straight channels. 

Page 4.

The section regarding the plasma-endothelium interface would be improved by further categorizing the topics as relating to “flow” or “transport”.  Line 163. The BBB is introduced without mention of its biological function, explaining why there are impediments to drug delivery. Line 165. Park was not the first to generate a BBB model from these 3 cell types as suggested here. See the work of Campisi et al., Biomaterials, 2018. Line 168. “Additional groups expanded…”. Include citations. Line 171. CNS is not spelled out the first time it is used.

Page 5.

Paragraph 2. The authors make EC mechanotransduction sound fully understood, but this is still an active area of research.  Line 220. “DIY” is a misnomer.  A more descriptive title would be something like “molded cylindrical EC monolayers” or similar.  Line 225. What does “adversely affected” mean? Reduced? Line 236. What are the “biophysical criteria” uncovered?

Page 6.

Line 246-247. There is more known about sprouting than the authors have implied.  Line 248-249. “Given that the role… angiogenic sprouting”. The authors’ conclusions are not supported by literature nor as obvious as they would have the readers believe.  Sprouting is often described as a strongly biochemical phenomenon with ECs responding to gradients of cytokines and chemokines, frequently driven by hypoxia-induced factors.  Furthermore, sprouts are dead-ended channels with minimal internal flow until the circuit is completed, so it is unclear how flow would “direct” them.  Line 253. This sentence is unclear and the authors seem to suggest that a VEGF concentration can induce interstitial flow whereas the flow was generated by pressure gradients.  The results of the paper in citation 70 are not well described.  Sentence beginning “Overall, these findings suggest…”. The authors seem to suggest that shear stress causes an EC to sprout and then interstitial pressure drives the direction of vessel formation. These conclusions are incorrect. Flow is not required for sprouting, and has been observed in many in vitro experiments in which endothelial cells sprout in response to chemical stimuli, not mechanical.  While mechanical stimuli have a role in many aspects of vascular biology, the chemo-sensitivity of ECs cannot be ignored.  Line 273-279 “Because of this shared… in response to shear stress”.  These sentences are unclear. 

Page 7

Line 315. What are the “multitude of roles” that erythrocytes play? Line 326. The authors seem to suggest that Tsvirkun developed a model of the glycocalyx, when in fact they are one of few groups to measure it and specifically examine it. Why is the cell-free layer 4.5 μm from the apical membrane if the glycocalyx thickness is only 600 nm? That thickness would occlude the smallest capillaries of 8-10 μm in diameter. 

Page 8 and 9.

Figure 3. What does this show?  Why is it important to the manuscript?  It does not confirm a 600 nm thickness of the glycocalyx, which would typically be imaged with electron microscopy. Line 346. Clarify whether the sialic acid residues are removed from the ECs or RBCs.  Could the discussion of malaria and sickle cell disease be subsections or reorganized for clarity? These paragraphs include much more detail than the preceding sections and might be better if made more concise. 

Page 10

Line 443-446. Also, cardiovascular disease.

Page 12.

Line 535. “Leukocyte stiffness plays a role” How? Citations? Paragraph beginning with “Expanding the capabilities…” There are similar studies in the literature using methods other than microfluidic devices. Line 560. What are “endothelium settings”? Paragraph beginning “Also integral to the pathogenesis of atherosclerosis…” Atherosclerosis has not been introduced in this text and the pathogenesis is not described clearly. Line 571. “Unpacking this seemingly important role” is too informal for a scientific text.

Page 13.

Line 584. The authors are uninformed of the use of endostatin in cancer. Though low in toxicity, it failed to show efficacy in many clinical trials.  Line 599. “angiogenic sprouting occurred directed toward the leukemic cell channel” is misleading because there was no other direction available for sprouting to occur.

Page 15.

Table 1. The absent/present dichotomy used in the table is not as useful as details. List specifics for each column when possible such as cell type or source, diameter (or range) of channels, type of geometry (straight, stenotic, branched, or complex in other ways?).  The final column is not superfluous, with only 1 study different from the rest.  Either change the criteria for “long-term” to include additional studies or remove this column and mention the difference of citation 100 in the text. 

Page 16.

The lymphatics are introduced, despite the titular focus on the “blood-endothelium interface”. Clarify that lymphatics conduct lymph, not blood. The “Perfused biofluids” section could be incorporated into other relevant sections rather than standing alone.

Page 18.

Line 745. Other hydrogels such as Matrigel or fibrin? Line 749 “scrapped” is too informal.

Reviewer 2 Report

This paper describes microfluidic methods to model the blood-endothelium interface. The authors show methods to develop microfluidic systems and contributions of these in vitro models. They also describe current limitations and challenges of the microfluidic models. Since this paper is well-written and contains useful information to develop vascularized microfluidic devices, I would recommend this paper should be accepted for publication in Micromachines.

The following should be revised:

- On line 627, the authors describe “we have for the first time been able to study the real-time cell-cell interactions…”. It is not clear what the authors want to show and which method this sentence indicates. The authors need clarify this sentence.

- Table 1 seems good to show; however, the authors do not mention about this table thoroughly. Since the authors describe current limitations and challenges in this section, the authors should mention how important each factor shown in Table 1 is for each interface. Because device design depends on purpose of study, the authors should mention all the factors to clarify which factors are required for each component of blood.

- The authors should double check abbreviations, e.g. DIY in line 220 and EGM in Figure 4 because they do not define them.

- There are two “4.5” sections. The section of “Future” should be a section of conclusion.

Round 2

Reviewer 1 Report

The authors have adequately addressed all of our previous concerns.

Author Response

Thank you.